# SOX2-OT Binds with ILF3 to Promote Head and Neck Cancer Progression by Modulating Crosstalk between STAT3 and TGF-β Signaling

**DOI:** 10.3390/cancers15245766

**Published:** 2023-12-08

**Authors:** Ru Wang, Yifan Yang, Lingwa Wang, Qian Shi, Hongzhi Ma, Shizhi He, Ling Feng, Jugao Fang

**Affiliations:** 1Department of Otorhinolaryngology, Head and Neck Surgery, Beijing Tong Ren Hospital, Capital Medical University, 1 Dongjiaominxiang Street, Beijing 100730, China; ruwang@mail.ccmu.edu.cn (R.W.); yangyifan517@ccmu.edu.cn (Y.Y.); 122019000294@ccmu.edu.cn (L.W.); cann-829@163.com (Q.S.); mahzent@126.com (H.M.); hsz_bjtr@mail.ccmu.edu.cn (S.H.); fengling@ccmu.edu.cn (L.F.); 2Key Laboratory of Otorhinolaryngology, Head and Neck Surgery, Beijing Institute of Otorhinolaryngology, Beijing 100730, China

**Keywords:** lncRNA SOX2-OT, head and neck squamous cell carcinoma, ILF3, STAT3

## Abstract

**Simple Summary:**

Head and neck squamous cell carcinoma (HNSCC) ranks seventh among malignant tumors worldwide, with an estimated 500,000 new cases annually. Despite the development of early diagnosis technology, there are still 60% of patients already in the middle–late stages when they were first diagnosed with HNSCC. Despite advances in multidisciplinary synthetic therapy, the overall survival rate remains low, with a five-year survival rate of less than 50%. Thus, it is urgently needed to search for novel early diagnosis biomarkers and therapy targets. In this study, we revealed the role of SOX2-OT in HNSCC progression and metastasis by binding with ILF3, which may serve as a therapeutic target and prognostic biomarker to improve the early diagnosis and overall survival of HNSCC.

**Abstract:**

Long non-coding RNA (lncRNA) is involved in the progression of head and neck squamous cell carcinoma (HNSCC). The molecular mechanism of lncRNA SOX2-OT in HNSCC remains unclear. Therefore, we aimed to elucidate the oncogenic role of SOX2-OT in HNSCC. QRT-PCR analysis was performed in 61 pairs of HNSCC cancer tissues, adjacent normal tissues, and 68 plasma samples confirmed that lncRNA SOX2-OT was overexpressed in cancer tissues and plasma samples, which served as a poor prognostic factor for HNSCC. The FISH assay demonstrated that SOX2-OT was localized in the nucleus and cytoplasm of HNSCC cell lines. Further, the cell function assay confirmed that SOX2-OT promoted cell proliferation and metastasis in vitro and in vivo. RNA pulldown and RIP assay results revealed that SOX2-OT bonds with ILF3 in HNSCC, and the rescue assay confirmed that SOX2-OT played an oncogenic role depending on ILF3 protein expression. Ingenuity pathway analysis and Western blotting indicated that SOX2-OT regulated HNSCC progression by promoting STAT3 phosphorylation and modulating the crosstalk between STAT3 and TGF-β signaling. These results reveal evidence for the role of SOX2-OT in HNSCC progression and metastasis by binding to ILF3, which may serve as a therapeutic target and prognostic biomarker in HNSCC.

## 1. Introduction

Head and neck squamous cell carcinoma (HNSCC) ranks seventh among malignant tumors worldwide, with an estimated 500,000 new cases annually [1]. Despite advances in surgery, radiotherapy, chemoradiotherapy, and immunotherapy, the overall survival rate remains low, with a five-year survival rate of less than 50% [2]. Unlimited proliferation and metastatic ability, typical hallmarks of cancer, are primarily responsible for the poor outcome of HNSCC.

Long non-coding RNAs (lncRNAs) have been defined as non-coding transcripts of more than 200 nucleotides (200 nt), which play complex and precise roles in cancer initiation and progression by acting as oncogenes or tumor suppressors. Evidence suggests that lncRNAs modulate the transcription and translation of functional core genes, serving as decoys, scaffolds, and competing endogenous RNAs (ceRNAs) [3]. Recently, abnormal expression of long non-coding RNAs (lncRNAs) has been reported in many cancers, including HNSCC [4], breast cancer [5], lung carcinoma [6], colorectal cancer [7], and esophageal cancer [8]. Various lncRNAs play roles in the progression, drug resistance, and metastasis of HNSCC and have identified lncRNAs as a novel set of potential prognostic factors and therapeutic targets in HNSCC [1]. lncRNAs contribute to the proliferation and metastasis of head and neck cancer by distinct mechanisms based on their subcellular localization and modulate or regulate the cancer immune microenvironment 11. For instance, lncMX1–215 regulates immunosuppression by disturbing GCN5/H3K27 acetylations and suppressing HNSCC proliferation and metastasis capacity, thus providing new strategies for immune checkpoint blockade treatment [9]. The lncRNA MIR31HG targets HIF1A/P21 to promote HNSCC tumorigenesis and proliferation by promoting cell cycle progression [10]. lncRNA POP1-1 promotes drug resistance in HNSCC by MCM5 and decelerates its degradation upon upregulation by VN1R59 [11]. Our previous study on lncRNA and mRNA integrated microarray demonstrated that lncRNA SOX2-OT was overexpressed in laryngeal squamous cell cancer (LSCC) tissues and played an oncogenic role in LSCC [12]. However, the function and regulation of SOX2-OT in HNSCC proliferation and metastasis are still unknown.

In this study, we monitored the expression of novel lncRNA, SOX2-OT, in HNSCC tissues. We also investigated its role in promoting cell proliferation and migration in vitro and in vivo and aimed to evaluate its action mechanism. Finally, we aimed to assess whether SOX2-OT can act as an oncogenic molecule in HNSCC and unravel the new regulatory mechanism in HNSCC tumor progression.

## 2. Materials and Methods

### 2.1. HNSCC Patients and Clinical Samples 

Human HNSCC and paired normal tissues were acquired from 57 patients who received surgery at Beijing Tongren Hospital. Additional fresh blood samples were obtained from 21 vocal cord polyps, 20 early-stage laryngeal SCC, and 27 advanced-stage laryngeal SCC. All the patients had signed informed consent. The research was approved by the Ethics Committee of Beijing Tongren Hospital. Histological grade was according to the World Health Organization. Tumor staging was according to the AJCC Cancer Staging Manual (7th edition). In this research, patients’ M stages were all M0.

### 2.2. RNA Isolation and Quantitative Real-Time Polymerase Chain Reaction (qRT-PCR)

Total RNA isolation was used by TRIzol reagent (Invitrogen, San Diego, CA, USA). Preparation of cDNA was used by MMLV Reverse Transcriptase (Fermentas, Toronto, ON, Canada). The SYBR Green fluorescence was measured by the ABI PRISM 7500 FAST system (Thermofisher, Waltham, MA, USA). The 2^−ΔΔCt^ method was used to perform the quantitation. Experiments were performed three times. The primers for PCR were as follows: SOX2-OT forward,5′-GGCTGGGAAGGACAGTTCG-3′; SOX2-OT reverse,3′-AGATGATCTTGCCAGGCGATC-5′; GAPDH forward,5′-GATCATCAGCAATGCCTCCT-3′; and GAPDH reverse,3′-TGAGTCCTTCCACGATACCA-5′.

### 2.3. Cell Culture and Transfection

HNSCC cell lines (TU686 and FaDu) were obtained from our laboratory. Dulbecco’s modified Eagle’s medium-high glucose (Gibco, Grand Island, NY, USA) was used to culture the cell lines. SiRNA (50 nM) transfection was performed by RNAiMAX (Invitrogen) according to the manufacturer’s protocol. The sequencing of siRNA was as follows: siSOX2-OT#1: 5′-UGGCAAGAUCAUCUCAAACUATT-3′; siSOX2-OT#2: 5′- GGAAGGACAGUUCGAGUUCAUTT-3′; and siCtrl: 5′-UUCUCCGAACGUGUCACGU-3′. The plasmids of overexpression-ILF3 (NM_001137673.2) were obtained from GeneChem Co. (Shanghai, China). pCDH-ILF3 plasmid was transfected by Lipofectamine™ 2000 (Invitrogen) according to the manufacturer’s protocol.

### 2.4. Cell Proliferation and Formation Assays

Cells were plated in 96-well plates after transfection (2000 cells/well). According to the manufacturer’s instructions, cell proliferation was observed every 24 h for four days. The CCK-8 solution (10 μL per well) (MedChemExpress, Shanghai, China) was mixed in a well and incubated at 37 °C for 1 h. A microplate reader (Multiscan FC, Thermo Scientific, Waltham, MA, USA) was used to count the formazan dye generated by absorbance at 450 nm. Then, the cells (500 cells/well) were placed in six-well plates and incubated for two weeks. A 0.1% crystal violet was used for the cell staining. A microscope was used for counting the positive colonies (containing >50 cells).

### 2.5. Flow Cytometry

SiSOX2-OT or control was used to transfect the FaDu and TU686 cells. Cells were stained with propidium iodide (PI) 48 h after the transfection. Then, we performed the flow cytometry. Briefly, cells were fixed with 70% ethanol (cold) dropwise and incubated at 4 °C for one night. PBS was used to wash the cells. Subsequently, the cells were thoroughly mixed with propidium iodide (5 μL) and binding buffer (500 μL) and incubated with fluorescein isothiocyanate-conjugated anti-annexin V antibody (5 μL). The results were evaluated using a CytoFLEX LX (Beckman Coulter, Miami, FL, USA). The data were analyzed using CytTExpert software (Version 2.4).

### 2.6. Western Blot Analysis

A 2 × SDS sample buffer was used for cell lysate preparation. Specific antibodies against ERP57 (Abcam Cambridge, MA. USA, ab13506, dilution rate, 1:1000), total STAT3 (Abcam, ab68153, dilution rate, 1:1000), p-STAT3 (Abcam, ab267373, dilution rate, 1:1000), SMAD3 (Abcam, ab40854, dilution rate, 1:1000), SMAD2 (Abcam, ab40855, dilution rate, 1:1000), TGF-β (Abcam, ab215715, dilution rate, 1:1000), ILF3 (Abcam, ab92355, dilution rate, 1:1000), β-actin (Abcam, ab8226, dilution rate, 1:1000), and GAPDH (Abcam, ab8245, dilution rate, 1:1000) were used for Western blot analysis. GAPDH was labeled as an internal reference. The secondary antibody anti-rabbit IgG (sc-2004, dilution rate, 1:10,000) and anti-mouse IgG (sc-2005, dilution rate, 1:10,000) were purchased from Santa Cruz (Santa Cruz, CA, USA). The ChemiDoc MP Imaging System (Bio-Rad, Hercules, CA, USA) was used for image acquisition.

### 2.7. Transwell Assay

Transwell chambers (Corning, Corning, NY, USA, 8-μm pores) coated with Matrigel (BD, San Jose, CA, USA) were used to perform the transwell assays. Briefly, cells were plated in the upper chambers (5 × 10^4^ cells) and incubated in serum-free medium (500 μL), whereas, in the lower chambers, 500 μL medium supplemented with 10% FBS was poured. Then, the cells were incubated in a 5% CO_2_ humidified incubator at 37 °C for 18 h, and the migrated cells were fixed with 1% formaldehyde. A 0.1% crystal violet was used to stain the cells. A light microscope (Olympus, Tokyo, Japan) was used for image capturing. For each filter, cells were counted in five random fields.

### 2.8. Mouse Xenograft Experiments

The Institutional Animal Care and Use Committee of the Capital Medical University approved all the experimental animal protocols. The shSOX2-OT (Vector No.: GV115) and shCtrl (Vector No.: GV115) lentivirus were purchased from GeneChem Co. (Shanghai, China). The sequencing of shRNA is as follows: siSOX2-OT#2: 5′- GGAAGGACAGUUCGAGUUCAUTT-3′. Twenty BALB/c nude mice (from 4 to 6-week-old females) were obtained from Beijing Vital River Laboratory Animal Technology Co., Ltd. (Beijing, China). Approximately 1 × 10^7^ FaDu cells stably knocked down for SOX2-OT were mixed with a 40% Matrigel matrix. The suspension of cells was then injected subcutaneously into nude mice. We measured the length and width of the mouse tumors every 2 days with calipers. Tumor volume was calculated using the following formula: tumor volume = a × b^2^ × 0.5. The tumors were then photographed and weighed. The mice were euthanized at the end of the experiment.

### 2.9. RNA Fluorescence In Situ Hybridization (FISH)

FaDu and TU686 cells were placed onto a glass sheet. A 4% paraformaldehyde was used to fix the cells (15 min). PBS was used to wash the cells, and 0.5% Triton X-100 was used for permeabilizing (10 min). The RNAscope Fluorescent Reagent Kit (ACDBio, Newark, CA, USA) was used to process the slides. The FISH probe mix of SOX2-OT was designed by ACDBio Co. After washing, the 4′, 6′-diamidino-2-phenylindole (DAPI) was used to counterstain the nuclear. A fluorescence microscope (Leica, Weztlar, Germany) was used to observe the cells.

### 2.10. RNA Pulldown Assay

Biotin pulldown analysis was used to detect the correlation between ILF3 protein and SOX2-OT mRNA. Biotin-labeled RNA was used to transfect FaDu cells. A 1% formaldehyde was used to cross-link with cells, and 0.125 M glycine was used to quench. The cells were resuspended in a lysis buffer on ice for 10 min and sonicated. The cell lysate was diluted twice with a hybridization buffer. Streptavidin Dynabeads (Thermo Fisher, Waltham, MA, USA) were blocked in lysis buffer containing 1 mg/mL yeast tRNA and 1 mg/mL BSA for 2 h at 4 °C and washed twice with 1 mL of lysis buffer. After adding 100μL of washed/blocked Dynabeads, the entire mix was rotated for 30 min at 37 °C. Beads were captured using magnets (Thermo Fisher, MA, USA) and washed five times. The beads were then subjected to RNA elution with buffer. The eluted supernatant was examined using mass spectrometry. Label-free quantification was used to analyze mass spectrometry data.

### 2.11. RNA Immunoprecipitation Assay

IP lysis buffer was lysed with FaDu cells (containing RNase inhibitor and protease inhibitor cocktail). Then, RNA immunoprecipitation (RIP) experiments were conducted using the anti-ILF3 antibody or control immunoglobulin G overnight at 4 °C. Then, the Dynabeads Protein G Immunoprecipitation Kit (Thermo Fisher, MA, USA) was incubated with the mixture according to the protocol. After washing and centrifugation with IP lysis buffer, the immunoprecipitated complex was collected. TRIzol reagent was used to isolate total RNA and mRNA expression of SOX2-OT was detected.

### 2.12. Statistical Analysis

Survival analysis was performed using Kaplan–Meier analysis with the log-rank test. The *t*-test was used to compare the means of the two groups. The One-Way ANOVA was applied to compare the means of more than two groups. Data were presented as mean ± standard deviation (SD). A *p* < 0.05 was set as the statistical significance. One star represented *p* < 0.05; two stars represented *p* < 0.01, and three stars represented *p* < 0.001. The statistical analysis was performed using the GraphPad Prism software (La Jolla, San Diego, CA, USA, version 7.0).

## 3. Results

### 3.1. SOX2-OT Overexpression Predicts Poor Overall Survival in HNSCC

The expression of SOX2-OT was measured in 57 HNSCC patients’ tissues and adjacent normal tissues by qRT-PCR. As shown in Figure 1A, SOX2-OT was high-expressed (*p* < 0.05) in HNSCC tissues. To study the expression level between the SOX2-OT and patients’ clinicopathological characteristics, according to the median SOX2-OT expression, we divided 57 patients into a high-SOX2-OT-expression group (*n* = 29) and a low-SOX2-OT-expression group (*n* = 28). As shown in Table 1, high SOX2-OT levels were associated with poor differentiation (*p* < 0.05). We further evaluated the expression of SOX2-OT in plasma samples, including 21 vocal cord polyps, 20 early-stage LSCC, and 27 advanced-stage LSCC. As shown in Figure 1B, SOX2-OT was significantly overexpressed in LSCC compared to polyps (*p* < 0.01).

To determine whether the expression of SOX2-OT could be a prognostic factor in HNSCC, from the Kaplan–Meier survival curve, we found that the OS rate was significantly better in the low-SOX2-OT group than in the high-SOX2-OT group (Figure 1C). Univariate Cox analysis demonstrated that high SOX2-OT levels were distinctly correlated with poor OS in patients with HNSCC. Further multivariate Cox analysis revealed that SOX2-OT overexpression influenced the lifetime of patients with HNSCC (*p* < 0.05) (Table 2). As the target gene of SOX2-OT, the expressions of SMAD2 and SMAD3 were positively correlated with STAT3 in TCGA-HNSCC data (Figure 1D,E).

### 3.2. SOX2-OT Promotes Cell Progression and Migration, Meanwhile Inhibits Apoptosis in HNSCC

To study the regulatory capacity of SOX2-OT, one control siRNA (siCtrl) and three siRNAs (siSOX2-OT#1, siSOX2-OT#2, and siSOX2-OT#3) were used. FaDu and TU686 cells were selected for lentivirus transfection. After the determination of qRT-PCR as the best knockdown efficiencies, siSOX2-OT#1 and # 2 were chosen for the following experiments (Figure 2A). According to the CCK-8 assays (Figure 2B) and the colony formation assays (Figure 2C), we found that after SOX2-OT knockdown, the cell proliferation of FaDu and TU686 cells was significantly decreased.

We further investigated the apoptosis using flow cytometry. In FaDu and TU686 cells, after SOX2-OT knockdown, the apoptotic cells increased (*p* < 0.05), indicating that SOX2-OT knockdown promoted cell apoptosis (Figure 2D). A transwell migration assay was performed to investigate the effect of SOX2-OT on the migration of HNSCC cells. These results demonstrated that migration of FaDu and TU686 cells in the transwell assay was reduced following SOX2-OT knockdown (Figure 2E), suggesting that SOX2-OT reduction inhibited HNSCC cell migration.

### 3.3. SOX2-OT Facilitates HNSCC Xenograft Growth

We further studied how SOX2-OT influenced the tumorigenesis in vivo. Firstly, we screened the stably transfected FaDu cells with siSOX2-OT or siCtrl; then, we subcutaneously injected the cells in the mice. Figure 3A shows the images of xenograft tumors. Compared to the control group, the weight and volume of the tumors in the SOX2-OT knockdown group were significantly reduced, suggesting that the knockdown of SOX2-OT may reduce the tumorigenesis of HNSCC in vivo (Figure 3B,C).

### 3.4. SOX2-OT Binds with ILF3 to Exert Function in HNSCC

The FISH results illustrated that SOX2-OT was abundant in the nucleus and cytoplasm (Figure 4A). However, the detailed regulatory mechanisms of SOX2-OT remain unclear. By using the sense or antisense strand of biotin-labeled SOX2-OT, we conducted RNA pulldown analysis and mass spectrometry to identify the interacting partners of SOX2-OT (Figure 4B). Mass spectrometry analysis identified 119 proteins (Appendix A). Among the RNA-binding proteins, ILF3 was the top binding protein with high coverage in FaDu cells. We then confirmed the relation between ILF3 and SOX2-OT by RIP assay, followed by qRT-PCR in a reciprocal experiment. SOX2-OT was gathered in FaDu cell immunoprecipitates significantly (Figure 4C). We explored the effect of ILF3 and SOX2-OT interactions in SOX2-OT-depleted HNSCC by Western blot analysis. We observed a significant change in the expression of the ILF3 protein upon SOX2-OT knockdown in FaDu cells (Figure 4D). Our findings clearly exhibit a direct and specific interaction between SOX2-OT and ILF3.

### 3.5. SOX2-OT Promoted HNSCC Cell Progression and Migration Depending on ILF3

To study the potential mechanism between SOX2-OT and ILF3, we used the TCGA database. According to the database, the positive relationships between SOX2-OT and ILF3 were observed in HNSCC tissues (Figure 5A). Furthermore, we explored whether SOX2-OT functions via ILF3. In FaDu and TU686 cells, SOX2-OT knockdown could reduce cell proliferation and migration, while the reduction could be rescued by overexpressing ILF3 simultaneously. In addition, silencing SOX2-OT mediated increase in apoptosis was partially rescued by simultaneous overexpression of ILF3 (Figure 5B–D). From the result, we found the impact of SOX2-OT on HNSCC is mediated by ILF3.

### 3.6. SOX2-OT Regulates the Crosstalk between STAT3 and TGF-β by ILF3 in HNSCC

We hypothesized that SOX2-OT regulates STAT3 and TGF-β signaling. We found that the activated STAT3 (phosphor-STAT3) was reduced significantly, and when the SOX2-OT was knocked down, the reduction in pSTAT3 could be rescued by overexpressing ILF3 in the SOX2-OT knockdown of FaDu and TU686 cell lines. It has been reported that ILF3 /ERp57 forms a positive feedback loop mediating STAT3; therefore, ILF3/STAT3/ERp57 together promote evident renal cell carcinoma proliferation. Therefore, we tested the expression of STAT3 and ERp57 after simultaneous knockdown of SOX2-OT and overexpression of ILF3 and observed that SOX2-OT played an oncogenic role in HNSCC by regulating ERp57 and STAT3 phosphorylation (Figure 6A). In addition, crosstalk exists between STAT3 signaling and Snail-Smad3/TGF-β1 in cancers. STAT3 activation exacerbates TGF-β1-induced EMT through Smad3/Snail signaling in cancers. In our study, we tested the expression of Smad2, Smad3, and TGF-β after the knockdown of SOX2-OT and ILF3 overexpression simultaneously. The results revealed that SOX2-OT regulation of the TGF-β pathway in HNSCC depends on ILF3 expression (Figure 6A and Appendix A). CO-IP assay validated that after SOX2-OT overexpression, the combination of STAT3 and SMAD3 was obviously increased (Figure 6B and Appendix A). In conclusion, we provide new insights that SOX2-OT binds to ILF3, playing an oncogenic role in HNSCC by modulating the interaction between STAT3 and TGF-β pathways.

## 4. Discussion

We previously demonstrated that lncRNA SOX2-OT was overexpressed in laryngeal squamous cell cancer (LSCC) tissues and played an oncogenic role in LSCC [12]. However, the potential regulatory mechanisms by which SOX2-OT promotes HNSCC cell proliferation and metastasis remain unclear. Therefore, in this study, we established that lncRNASOX2-OT promotes HNSCC cell proliferation and metastasis. Mechanistically, we confirmed that by directly binding with ILF3, SOX2-OT promoted STAT3 phosphorylation and activated the TGF-β signaling pathway. We also demonstrated that SOX2-OT overexpression was associated with worse clinical outcomes in HNSCC patients. Thus, SOX2-OT functions as an oncogenic lncRNA and plays a pro-metastatic role in HNSCC, highlighting a novel regulatory mechanism underlying HNSCC tumor progression.

In the past five years, aberrant expression of the lncRNA SOX2-OT has been reported in some cancers, including breast cancer [13], non-small cell lung cancer [14], ovarian cancer [15], hepatocellular carcinoma [16], nasopharyngeal carcinoma [17], and glioblastoma [14]. SOX2-OT overexpression was significantly associated with worsening clinical outcomes, suggesting the potential prognostic value of SOX2-OT [14]. In our study, SOX2-OT was overexpressed in HNSCC cancer tissues and plasma samples and served as a poor prognostic factor for HNSCC patients. SOX2-OT promotes carcinogenesis and tumor metastasis and acts as an oncogene in most cancers [18]. In accordance with the literature, our results confirmed that SOX2-OT promoted cell proliferation and migration both in vitro and in vivo, thus playing a carcinogenic role in HNSCC.

Recently, numerous lncRNAs have played crucial roles in multiple biological processes, including transcriptional regulation, epigenetic alterations, sequestration of miRNAs or proteins, and RNA or protein modification [19]. LncRNAs perform these functions via a variety of different mechanisms based on their subcellular localization, including acting as “scaffold” guides for chromatin-modifying enzymes, as in the case of HOTAIR or DLX6AS [20,21], serving as molecular signals, as in the case of Linc-p21 or PANDA [22,23], acting as competing endogenous RNAs that “sponge” microRNAs or proteins [24] and facilitating or inhibiting RNA/protein modification [25]. According to our results, SOX2-OT was localized in both the nucleus and cytoplasm in HNSCC. The mechanistic study suggested that SOX2-OT directly binds to ILF3, and the rescue assay confirmed that SOX2-OT played an oncogenic role depending on ILF3 protein expression in HNSCC. The role of ILF3 in cancer biology is definite, and it participates in diverse cellular functions such as mRNA stabilization, translation inhibition, and non-coding RNA biogenesis [26]. ILF3 regulates VEGF mRNA stability in breast cancer and modulates cyclin E1 mRNA stability in hepatocellular carcinoma [27,28]. However, we did not observe ILF3-mediated SOX2-OT stabilization but their direct binding to HNSCC cells.

The activation of STAT3 and TGF-β signaling is frequently detected in human cancers and is implicated in the proliferation and metastasis of various cancers. STAT3 is a member of the STAT family, which regulates various target genes with critical functions in tumorigenesis, including cell proliferation, apoptosis, inflammation, and angiogenesis. It has been reported that ILF3/ERp57 forms a positive feedback loop mediating STAT3; therefore, ILF3/STAT3/ERp57 together promote clear cell renal cell carcinoma proliferation [29]. Consequently, we evaluated the expressions of STAT3 and ERp57 after simultaneous knockdown of SOX2-OT and overexpression of ILF3 and observed that SOX2-OT played an oncogenic role in HNSCC by regulating ERp57 and phosphorylation of STAT3. In addition, there is tight crosstalk between STAT3 and TGF-β signaling in various cancers, including gastrointestinal cancer [30], lung cancer [31], and hepatocellular carcinoma [32]. TGF-β1-induced EMT depends on crosstalk with STAT3 signaling in hepatocellular carcinoma (HCC) cells, and positive p-STAT3 and TGF-β1 proteins were co-expressed in the liver tissues of HCC patients [32]. Inhibition of STAT3 signaling can relieve disease through crosstalk with Snail-Smad3/TGF-β1 in both human and rat HCC. In this study, we tested the expression of Smad2, Smad3, and TGF-β after simultaneous knockdown of SOX2-OT and ILF3 overexpression. The results confirmed that SOX2-OT regulation of the TGF-β pathway in HNSCC depends on ILF3 expression. Finally, we demonstrated that SOX2-OT binds to ILF3 to induce proliferation and metastasis in HNSCC by modulating the interaction between the STAT3 and TGF-β pathways.

## 5. Conclusions

We discovered the oncogenic role of SOX2-OT binding with ILF3, which promotes cell proliferation and migration by regulating STAT3 and TGF-β signaling in HNSCC. Further, SOX2-OT is a poor prognostic factor for HNSCC. Our work provides a promising prognostic and therapeutic target for the treatment of HNSCC.

## Figures and Tables

**Figure 1 cancers-15-05766-f001:**
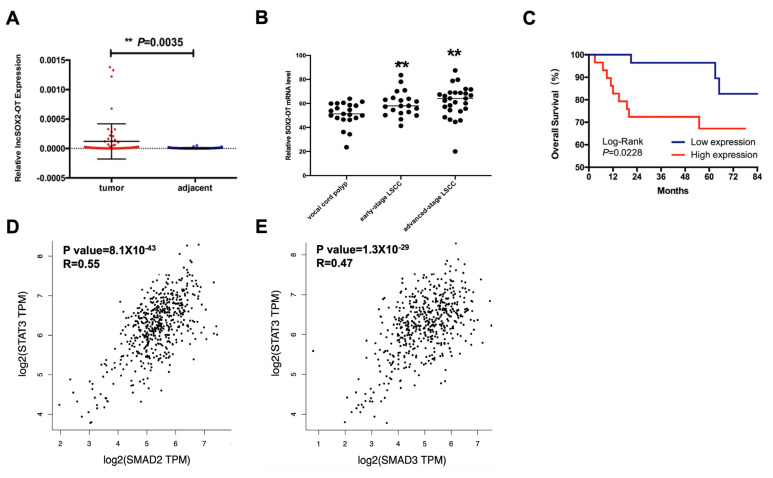
SOX2-OT overexpression in HNSCC predicts poor overall survival. (**A**) SOX2-OT was highly expressed (*p* < 0.05) in 57 HNSCC tissues. (**B**) SOX2-OT was significantly overexpressed in 47 LSCC plasma compared to 21 polyp plasma (*p* < 0.01). (**C**) The OS rate was significantly better in the low-SOX2-OT group than in the high-SOX2-OT group. The expression of SMAD2 (**D**) and SMAD3 (**E**) were positively correlated with STAT3, according to TCGA-HNSCC data. ** *p* < 0.01. All experiments were repeated at least three times.

**Figure 2 cancers-15-05766-f002:**
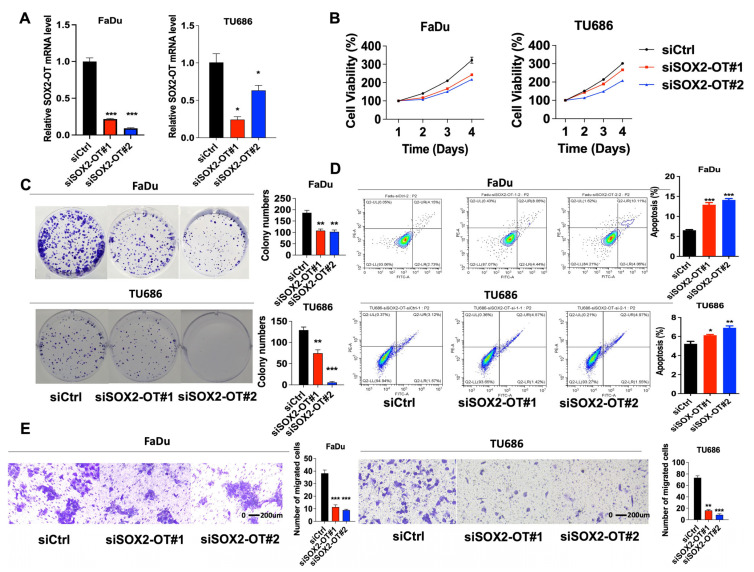
SOX2-OT regulated HNSCC cell growth, apoptosis, and migration. (**A**) FaDu and TU686 cells were selected for siRNA transfection. After 48 h transfection, total mRNA of cells was isolated, and qRT-PCR was performed to detect the knockdown efficiencies of siSOX2-OT. (**B**) After 48 h transfection, cells were seeded to 96 wells, and the CCK-8 assay was performed on Day 1, Day 2, Day 3, and Day 4. (**C**) After 48 h transfection, cells were seeded to 6 wells, and the colony formation assay was performed. (**D**) Cell apoptosis was tested at 48 h after transfection. The results showed that SOX2-OT knockdown promoted cell apoptosis. (**E**) Transwell was tested at 48 h after transfection, and migration of FaDu and TU686 cells in the transwell assay was reduced following SOX2-OT knockdown. * *p* < 0.05; ** *p* < 0.01; *** *p* < 0.001. All experiments were repeated at least three times.

**Figure 3 cancers-15-05766-f003:**
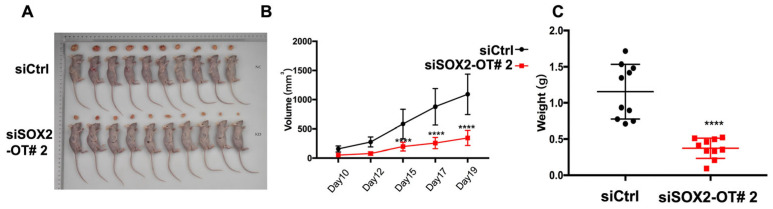
Knockdown of SOX2-OT inhibited HNSCC xenograft growth. (**A**) The images of tumors. Compared to the control group, the weight (**B**) and volume (**C**) of the tumors in the SOX2-OT knockdown group were significantly reduced. **** *p* < 0.0001.

**Figure 4 cancers-15-05766-f004:**
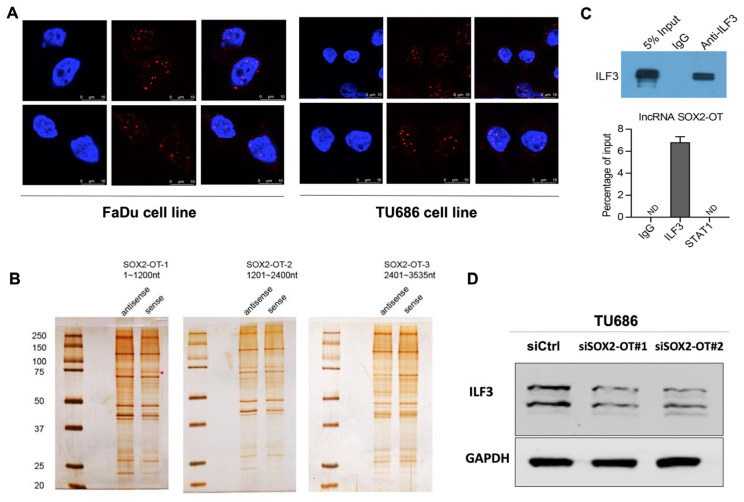
SOX2-OT bonded with ILF3 in HNSCC cells. (**A**) FISH results illustrated that SOX2-OT was abundant in the nucleus and cytoplasm. (**B**) RNA pulldown analysis and mass spectrometry to identify the interacting partners of SOX2-OT. (**C**) RIP assay followed by qRT-PCR confirmed the combination between ILF3 and SOX2-OT. (**D**) Western blot assay validated a significant change in the expression of the ILF3 protein upon SOX2-OT knockdown in TU686 cells.

**Figure 5 cancers-15-05766-f005:**
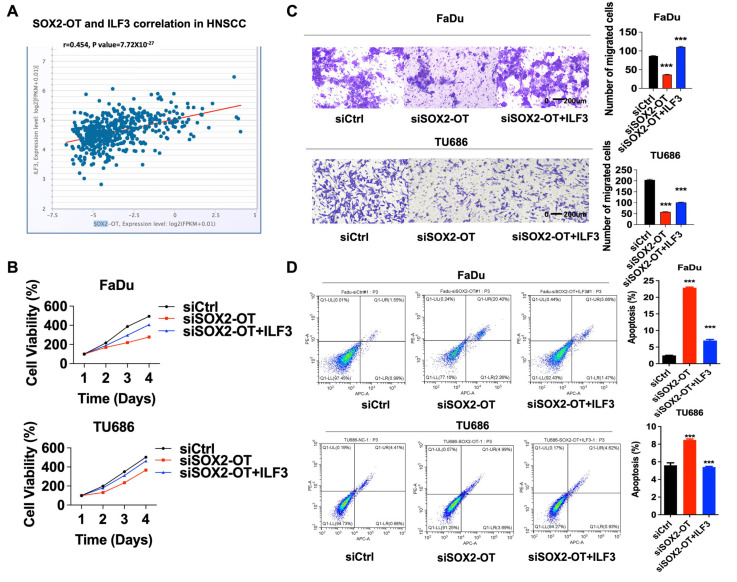
ILF3 mediated the effect of SOX2-OT on HNSCC cell progression and migration. (**A**) the positive relationship between SOX2-OT and ILF3 was observed in HNSCC tissues. In FaDu and TU686 cells, SOX2-OT knockdown could reduce the cell proliferation (**B**) and migration (**C**), while the reduction could be rescued by overexpressing ILF3 simultaneously compared with siSOX2-OT group. (**D**) In addition, silencing SOX2-OT mediated increase in apoptosis was partially rescued by simultaneous overexpression of ILF3 compared with siSOX2-OT group.Red: blank control, Green: negative control, Blue: laboratory group. *** *p* < 0.001. All experiments were repeated at least three times.

**Figure 6 cancers-15-05766-f006:**
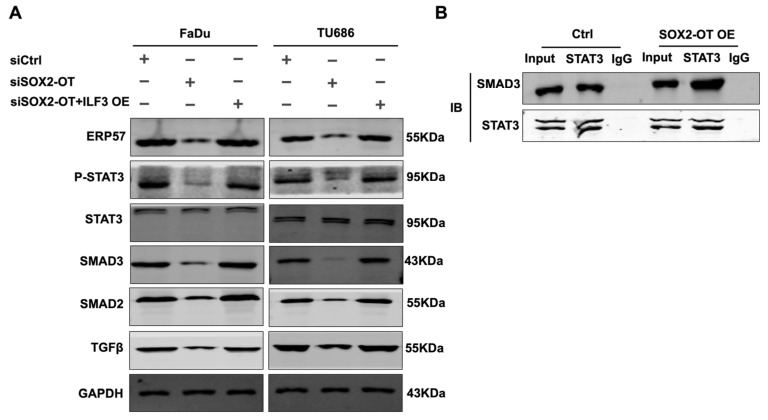
SOX2-OT modulated the interaction between the STAT3 and TGF-β pathways by ILF3 in HNSCC. (**A**) Western blot of ERP57, P-STAT3, STAT3, SMAD3, SMAD2, and TGFβ protein expression after SOX2-OT knockdown with or without ILF3 overexpression in HNSCC cells. (**B**) Immunoprecipitation of SMAD3 by anti-STAT3 performed in HNSCC cells after SOX2-OT overexpression.

**Table 1 cancers-15-05766-t001:** Clinicopathological characteristics of 57 HNSCC patients.

Characteristics	Cases	SOX2-OT Relative Expression		*p*-Value
		Low	High	
Age years				0.301
<60	28	16	12	
≥60	29	14	15	
T category				0.705
T1–T2	34	16	18	
T3–T4	23	12	11	
Lymph-node metastasis				0.698
N0	44	21	23	
N1–N2	13	7	6	
Clinical stage				0.903
I–II	31	15	16	
III–IV	26	13	13	
Differentiation				0.031
Well	21	15	6	
Moderate	32	11	21	
Poor	4	2	2	

**Table 2 cancers-15-05766-t002:** Univariate and multivariate COX analysis of clinical pathological feature for OS in HNSCC patients.

Variable	Univariate	Multivariate		*p*-Value
		Hazard Ratio	95% CI	
Age years				
<60 vs. ≥60	0.302			
T category				
T1–T2 vs. T3–T4	0.644			
Lymph-node metastasis				
N0 vs. N1–N2	0.769			
Clinical stage				
I–II vs. III–IV	0.989			
Differentiation				
Well vs. Moderate–Poor	0.087	2.507	0.716–8.782	0.151
SOX2-OT expression				
High vs. Low	0.023	3.811	1.006–14.444	0.049

## Data Availability

The datasets used and/or analyzed during the current study are available from https://doi.org/10.6084/m9.figshare.24454147 (accessed on 1 January 2023).

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
