# Peer review of "SOX2-OT Binds with ILF3 to Promote Head and Neck Cancer Progression by Modulating Crosstalk between STAT3 and TGF-β Signaling"

_cancers, 2023, doi:10.3390/cancers15245766_

Round 1

Reviewer 1 Report

Comments and Suggestions for Authors

SOX2-OT binds with ILF3 to promote head and neck cancer progression by modulating crosstalk between STAT3 and TGF-β signaling

The authors in the manuscript have shown that 1) lncRNA SOX2-OT is over expressed in Head and Neck squamous cell carcinoma (HNSCC) with over expressors having significantly low overall survival rate. 2) Demonstrated the oncogenic potential of lncRNA SOX2-OT using in vitro squamous carcinoma cells (FaDu and TU686) and in vivo xenograft model utilizing FaDU and FaDU cells with stably transfected siRNA for lncRNA SOX2-OT 3) Demonstrated the positive co-relation of lncRNA SOX2-OT and ILF3 levels in HNSCC tissues. 4) Demonstrated the proposed mechanism that lncRNA SOX2-OT knock down also reduces ILF3 protein levels. Over expressing ILF3 rescues the oncogenesis in these cell lines.  Further, decrease in lncRNA SOX2-OT with siRNA also decreases the phosphorylation of STAT3, SMADS and TGFβ protein levels. Also, overexpression of lncRNA SOX2-OT increases the cross talk of STAT3 and SMAD cross talk by facilitating their association in vitro.

Comments:

1.       Since ILF3 regulates oncogenesis is it possible that ILF3 alone would be sufficient to cause carcinogenesis. Because when the authors overexpress ILF3 levels in lncRNA SOX2-OT knock out cells, ILF3 by itself can promote oncogenesis. Have the authors tried over expressing lncRNA SOX2-OT or ILF3 in normal non-cancerous cell lines?

2.       Have the authors investigated the association of SMADS and STAT3 in HNSCC tissues.

3.       It would be helpful if the authors also briefly introduced to the clinical stages, T categories, N categories to help readers from other scientific discipline.

Introduction: It would be helpful if the authors provided the full forms of the abbreviations. Most of the abbreviations: GCN5, H3K27, HIF1A, POP1 and further in the text ILF3, ERp57 are not explained.

Materials and Methods:

2.3: Cell culture and transfection: Pleases provide the concentration of the and details of the siRNA, plasmid vectors that were used in the study. The details of lncRNA SOX2-OT overexpression vector not included here.

2.6: Western blot analysis: Please provide the concentration of primary antibodies used, details of secondary antibody. Details of ILF3 antibody not provided.

2.8: Mouse xenograft experiments. Please provide the number, age, gender of the mice used in the study.

Results:

Figures: All figures: The size of the figures, axes titles can be increased for better visibility.

Figure 1: It would be helpful to include the number of patients in each group in the legend as well.

Figure 2: No explanation of the observed results with lncRNA SOX2-OT siRNA3 in TU686 cells. Please include the time post- treatment at which the data was collected and number of times the experiments were repeated and P (significance value) in the legend.

Figure 3: Panel A. Please label the groups. What was the time point when these samples were collected. Please indicate the number of mice per group and the p value in the legend.

Figure 4: Please indicated the magnification of the images. In panel C there is no explanation for what was used here as anti-ILF3 (antibody/ siRNA?). No explanation of STAT1 group in the methods, results, legend or discussion. Panel D. Please indicate the molecular weight of the ILF3 protein. Why are there two distinct bands. Please explain what NC is. Why are two endogenous controls used: Actin and GAPDH. No details provided about the actin antibody in the methods section.

3.5: Line 259. What is the TCGA data base?

Figure 5: Panel C and D: Please check the labels of images. Please provide the time point the data was collected, the number of times the experiment was repeated and p value in the legend.

Figure 6: Please indicate the molecular weights of the proteins. B. Please include details of the SOX2-OT overexpression vector, concentration and additional details including the duration of experiment and reproducibility.

Discussion: Discussion can be improved significantly. It is not clear how ILF3 interacts with STAT3 and SMADS. It is not clear how ERP57 is related to this study. The last paragraph in the discussion has no citations/references.

Comments on the Quality of English Language

Can be improved

Author Response

Comments:

Since ILF3 regulates oncogenesis is it possible that ILF3 alone would be sufficient to cause carcinogenesis. Because when the authors overexpress ILF3 levels in lncRNA SOX2-OT knock out cells, ILF3 by itself can promote oncogenesis. Have the authors tried over expressing lncRNA SOX2-OT or ILF3 in normal non-cancerous cell lines?

Reply: Thank you for your detailed and constructive feedback. We will conduct additional experiments overexpressing ILF3 in normal non-cancerous cell lines in the future, providing a more comprehensive understanding of the role of ILF3 in oncogenesis.

Have the authors investigated the association of SMADS and STAT3 in HNSCC tissues.

Reply: Thanks for this comment. We appreciate the suggestion to investigate the association of SMADs and STAT3 in HNSCC tissues. We verified the figure 1 addressing this point using The Cancer Genome Atlas (TCGA)-HNSCC data. The results showed the expression of SMADS were positively correlated with STAT3, shedding light on the potential crosstalk between these signaling pathways in HNSCC.

It would be helpful if the authors also briefly introduced to the clinical stages, T categories, N categories to help readers from other scientific discipline.

Reply: Thanks for your suggestion. Doctors use the TNM staging system for most types of cancer. The TNM system uses letters and numbers to describe the tumor (T categories), lymph nodes (N categories), whether the cancer has spread or metastases (M categories). 

Introduction: It would be helpful if the authors provided the full forms of the abbreviations. Most of the abbreviations: GCN5, H3K27, HIF1A, POP1 and further in the text ILF3, ERp57 are not explained.

Reply: Thanks for your suggestion. The abbreviations were supplemented in the manuscript as follow:

GCN5: General control non-depressible 5

H3K27: lysine 27 on histone H3

HIF1A: Hypoxia Inducible Factor 1 Subunit Alpha

POP1: Processing of Precursor 1 

ILF3:  Interleukin enhancer-binding factor 3

ERp57: Protein Disulfide Isomerase Family A Member 3

Materials and Methods:

2.3: Cell culture and transfection: Pleases provide the concentration of the and details of the siRNA, plasmid vectors that were used in the study. The details of lncRNA SOX2-OT overexpression vector not included here.

Reply: Thanks for this comment. We provided the information on the “2.3. Cell culture and transfection” section as followed: SiRNA (50 nM) transfection was performed by RNAiMAX (Invitrogen) according to the manufacturers’ protocol. The sequencing of siRNA as followed: siSOX2-OT#1: 5’-UGGCAAGAUCAUCUCAAACUATT-3’, siSOX2-OT#2: 5’- GGAAGGACAGUUCGAGUUCAUTT-3’, siSOX2-OT#23: 5’- GCAAAUAAGAUAGGCAUACUUTT-3’, siCtrl: 5’-UUCUCCGAACGUGUCACGU-3’. The plasmids of overexpression-ILF3 (NM_001137673.2) were got from GeneChem Co. (Shanghai, China). pCDH-ILF3 plasmid was transfected by Lipofectamine™ 2000 (Invitrogen) according to the manufacturer’s protocol.

2.6: Western blot analysis: Please provide the concentration of primary antibodies used, details of secondary antibody. Details of ILF3 antibody not provided.

Reply: Thanks for this comment. We provided the information on the “2.6. Western blot analysis” section as followed: A 2×SDS sample buffer was used for cell lysates preparing. Specific antibodies against ERP57 (Abcam, ab13506, dilution rate, 1:1000), total STAT3 (Abcam, ab68153, di-lution rate, 1:1000), p-STAT3 (Abcam, ab267373, dilution rate, 1:1000), SMAD3 (Abcam, ab40854, dilution rate, 1:1000), SMAD2 (Abcam, ab40855, dilution rate, 1:1000), TGF-β (Abcam, ab215715, dilution rate, 1:1000), ILF3 (Abcam, ab92355, dilution rate, 1:1000) and GAPDH (Abcam, ab8245, dilution rate, 1:1000) were used for western blot analysis. GAPDH was labeled as an internal reference. The secondary antibody anti-Rabbit IgG (sc-2004, dilution rate, 1:10000) and anti-Mouse IgG (sc-2005, dilution rate, 1:10000) was purchased from Santa-Cruz. ChemiDoc MP Imaging System (Bio-Rad, USA) was used for the image acquiring.

2.8: Mouse xenograft experiments. Please provide the number, age, gender of the mice used in the study.

Reply: Thanks for this comment. Number: 20; Age:4 weeks; Gender: female.

Results:

Figures: All figures: The size of the figures, axes titles can be increased for better visibility.

Reply: Thanks for your suggestion. We increased the figure size in the manuscript.

Figure 1: It would be helpful to include the number of patients in each group in the legend as well.

Reply: Thanks for your suggestion. We included the number of patients in each group in the legend.

Figure 2: No explanation of the observed results with lncRNA SOX2-OT siRNA3 in TU686 cells. Please include the time post- treatment at which the data was collected and number of times the experiments were repeated and P (significance value) in the legend.

Reply: Thanks for this comment. After the determination of qRT-PCR, as the best knockdown efficiencies, siSOX2-OT#1 and # 2 were chosen for the following experiments, we excluded lncRNA SOX2-OT siRNA3 for further study. Three times the experiment was repeated. The P value was indicated by star on the top of the bar. One star: P<0.05, Two stars: P<0.01, Three stars: P<0.001.

Figure 3: Panel A. Please label the groups. What was the time point when these samples were collected. Please indicate the number of mice per group and the p value in the legend.

Reply: Thanks for this comment. The label was supplemented as suggested. The time point was horizontal coordinate 12 days, 15days, 17days, and 19 days. We used * represent P value in the figure. * P<0.05, ** P<0.01, *** P<0.001.

Figure 4: Please indicated the magnification of the images. In panel C there is no explanation for what was used here as anti-ILF3 (antibody/ siRNA?). No explanation of STAT1 group in the methods, results, legend or discussion. Panel D. Please indicate the molecular weight of the ILF3 protein. Why are there two distinct bands. Please explain what NC is. Why are two endogenous controls used: Actin and GAPDH. No details provided about the actin antibody in the methods section.

Reply: Thanks for this comment. The antibody was used for anti-ILF3. The STAT1 was not detected, so it is not significant. The predicated molecular weight of the ILF3 protein is 95kDa.Actually, we detected two bands of ILF3. It may cause by different transcripts in the cells. We changed the NC to siCtrl, and we also changed the KD1 and KD2 to siSOX2-OT1 and siSOX2-OT2, respectively. There are two loading control, so we deleted actin.

3.5: Line 259. What is the TCGA data base?

Reply: Thanks for this comment. The Cancer Genome Atlas (TCGA) database molecularly characterized over 20,000 primary cancer and matched normal samples spanning 33 cancer types. The over 2.5 petabytes of data generated through TCGA remain publicly available for anyone in the research community to use.

Figure 5: Panel C and D: Please check the labels of images. Please provide the time point the data was collected, the number of times the experiment was repeated and p value in the legend.

Reply: Thanks for this comment. We provided the time point the data was collected in Figure 2 legend. The Figure 5 experiments were same as in Figure 2. The number of times the experiment was repeated: Three. One star: P<0.05, Two stars: P<0.01, Three stars: P<0.001.

Figure 6: Please indicate the molecular weights of the proteins. B. Please include details of the SOX2-OT overexpression vector, concentration and additional details including the duration of experiment and reproducibility.

Reply: Thanks for this comment. We have included the molecular weights of the proteins. The additional experimental details were supplemented.

Discussion: Discussion can be improved significantly. It is not clear how ILF3 interacts with STAT3 and SMADS. It is not clear how ERP57 is related to this study. The last paragraph in the discussion has no citations/references.

Reply: Thanks for this comment. We improved and supplemented references in discussion as suggested. It has been reported that ILF3/ERp57 forms a positive feedback loop mediating STAT3; therefore, ILF3/STAT3/ERp57 together promote clear cell renal cell carcinoma proliferation. So, we included ERp57 in our study. The direct and specific mechanism of how ILF3 interacts with STAT3 and SMADS still needs further study, we will continue this study.

Reviewer 2 Report

Comments and Suggestions for Authors

In the manuscript entitled “SOX2-OT binds with ILF3 to promote head and neck cancer progression by modulating crosstalk between STAT3 and TGF-β signaling”, Wang and colleagues provided insights into the role of SOX2-OT in HNSCC progression and metastasis and its relationship with ILF3. Furthermore, they investigated the regulation of STAT3 and TGF-β signaling pathway by SOX2-OT. This work offered interesting findings for potential targeting lncRNA SOX2-OT as a therapeutic approach to head and neck cancer. The opinions will undoubtedly be of importance to the field.

However, I have listed below a series of suggestions which would further improve the quality of the manuscript. The questions and concerns must be addressed before the publication of this paper.

1. The resolution of the figures is low, and some figures are too small. The unclarity of some main figures will make the readers hard to tell the significant stars (e.g., Figure 1B), annotations (e.g., Figure 2D: It’s unable to see the percentage readout of FACS), or foci staining (e.g., Figure 4A). Could the authors generally improve all the figures to make them clearer?

2. The method parts lack some important information. The authors should include the sequences of used SOX2-OT-siRNAs in 2.3 and 2.8. In 2.5, how to stain for apoptosis? In 2.6, the authors should list the catalogue number of used antibodies. In 2.12, the authors should also annotate what did the stars indicate for. Please describe the methods thoroughly.

3. In Figure 2B TU686, the siSOX2-OT#1 didn’t show much decrease. Is it significant? Also, there are no stars in Figure 2C above. Please add the significance or explain.

4. In Figure 3, which row of xenograft tumors indicate for NC or KD? What do the NC and KD indicate for? Also, which siRNA (#1 or #2) is used in this figure as well as the following figures? Please annotate them.

5. What cell lines are used in Figure 4D? TU686 in the figure or FaDu mentioned in the legends?

6. In Figures 5C and 5D, the annotations are wrong. There are no #2, which should be “+ILF3”. Also, the significant stars are confusing. Should the authors compare the groups -/+ ILF3? Please correct.

7. In Figures 5 and 6, the authors must overexpress the ILF3 also in the siCtrl background and see the results. This is an important control. Is it possible that ILF3 could generate the phenotypes independent of SOX2-OT knockdown?

8. The authors should be more careful about the details. The problems include but are not limited to: In Figure 2. Fadu or FaDu?

Line 224, “migration” should be “Migration”.

In Figure 6A, it’s better to locate the bands in the central of each image.

Author Response

Comments:

  1. The resolution of the figures is low, and some figures are too small. The unclarity of some main figures will make the readers hard to tell the significant stars (e.g., Figure 1B), annotations (e.g., Figure 2D: It’s unable to see the percentage readout of FACS), or foci staining (e.g., Figure 4A). Could the authors generally improve all the figures to make them clearer?

Reply: Thanks for this comment. We acknowledge the concern about figure resolution and clarity. In response, we have improved the resolution of all figures and increased the size of relevant details (e.g., stars, annotations, and foci staining). This enhancement aims to ensure that readers can clearly interpret the figures.

  1. The method parts lack some important information. The authors should include the sequences of used SOX2-OT-siRNAs in 2.3 and 2.8. In 2.5, how to stain for apoptosis? In 2.6, the authors should list the catalogue number of used antibodies. In 2.12, the authors should also annotate what did the stars indicate for. Please describe the methods thoroughly.

Reply: Thanks for this comment. We provided the sequences of used SOX2-OT-siRNAs in 2.3 and 2.8,the information of apoptosis staining in 2.5, the catalogue number of used antibodies in 2.6, and the stars indication in 2.12.

  1. In Figure 2B TU686, the siSOX2-OT#1 didn’t show much decrease. Is it significant? Also, there are no stars in Figure 2C above. Please add the significance or explain.

Reply: Thanks for this comment. It is significant in Figure 2B. We add the significance stars above the bar.

  1. In Figure 3, which row of xenograft tumors indicate for NC or KD? What do the NC and KD indicate for? Also, which siRNA (#1 or #2) is used in this figure as well as the following figures? Please annotate them.

Reply: Thanks for this comment.  First row: NC; control group. Second row: KD; knockdown of SOX2-OT. siRNA#2 is used in this figure as well as the following figures because of the better knockdown efficiency. We revised Figure 3 as suggested.

  1. What cell lines are used in Figure 4D? TU686 in the figure or FaDu mentioned in the legends?

 Reply: Thanks for this comment. I am very sorry for the mistake, TU686 in the figure.

  1. In Figures 5C and 5D, the annotations are wrong. There are no #2, which should be “+ILF3”. Also, the significant stars are confusing. Should the authors compare the groups -/+ ILF3? Please correct.

Reply: Thanks for this comment. We have corrected the annotations in Figures 5C and 5D, providing the appropriate labels and addressing the issue of confusing significant stars.

  1. In Figures 5 and 6, the authors must overexpress the ILF3 also in the siCtrl background and see the results. This is an important control. Is it possible that ILF3 could generate the phenotypes independent of SOX2-OT knockdown?

Reply: Thanks for this comment. We appreciate the suggestion to overexpress ILF3 in the siCtrl background. We will conduct additional experiments in the future. These experiments help address the possibility that ILF3 could generate phenotypes independent of SOX2-OT knockdown.

  1. The authors should be more careful about the details. The problems include but are not limited to: In Figure 2. Fadu or FaDu?

Line 224, “migration” should be “Migration”.

In Figure 6A, it’s better to locate the bands in the central of each image.

Reply: Thanks for this comment.  We have addressed specific details, correcting "Fadu" to "FaDu," fixing the capitalization of "Migration" in line 224.

We believe that these revisions enhance the clarity, completeness, and accuracy of the manuscript. We remain open to further suggestions and appreciate your valuable feedback.

Reviewer 3 Report

Comments and Suggestions for Authors

The article describes a very important problem related to cancer, which is considered a disease of civilization. Which highlights the importance of the topic. Head and neck cancers are a serious problem, as their detection rate increased by over 25% last year. The authors clearly and clearly outline the problem in the introduction of the work. The results are presented in a substantive and clear way. Various research methods were used. My only complaint could be that the bibliography is too short, although it is relatively up-to-date. I believe that the article is suitable for publication

Author Response

We are grateful for your overall positive assessment and belief that the article is suitable for publication. We acknowledge the importance of a comprehensive bibliography, we aimed to focus on the most relevant and recent literature to maintain the article's conciseness, and supplemented related references as suggested.

Round 2

Reviewer 2 Report

Comments and Suggestions for Authors

The authors have largely responded and addressed the major concerns. The manuscript quality was significantly improved. 

Here, I still have some suggestions for further revising the manuscript. When solved, I have no hesitation to suggest the publication of this paper.

Please still double check the new revisions including but not limit to:

Figure 4 legends: “Tu686” should be “TU686”.

In Figures 5C and 5D, the authors should still indicate for which groups the significant stars are compared. Are the +ILF3 groups significant compared with -ILF3 groups or with siCtrl? Please clarify in legends or in manuscript.

Author Response

We appreciate your insightful and detailed comments and suggestions. Here are our responses to each of your points:

Comments:

Please still double check the new revisions including but not limit to:

Figure 4 legends: “Tu686” should be “TU686”.

Reply: Thank you for your detailed feedback. We revised the legends “Tu686” into “TU686” in Figure 4, and double checked the whole manuscript.

In Figures 5C and 5D, the authors should still indicate for which groups the significant stars are compared. Are the +ILF3 groups significant compared with -ILF3 groups or with siCtrl? Please clarify in legends or in manuscript.

Reply: Thanks for this comment. The +ILF3 groups significant compared with -ILF3 groups. We clarified the comparison in figure 5 legend as suggested. In FaDu and TU686 cells, SOX2-OT knockdown could reduce the cell proliferation and migration while the reduction could be rescued by overexpressing of ILF3 simultaneously compared with siSOX2-OT group. In addition, silencing SOX2-OT mediated increase in apoptosis was partially rescued by simultaneous overexpression of ILF3 compared with siSOX2-OT group.